# Learning Causal Structures from Mixed Dynamics via Polynomial Chaos Expansion

## Abstract

Real-world systems are rarely purely linear or nonlinear, but are instead a complex mixture of both. This heterogeneity makes them exceedingly difficult for standard causal discovery algorithms, which are typically designed for one regime and are brittle when applied to the other. Linear models miss critical nonlinear effects, while general nonlinear methods are computationally expensive and notoriously prone to discovering spurious relationships. We propose a new framework that robustly learns causal structures from such mixed-dynamics systems by learning from a spectral representation of model residuals. Our approach first identifies a sparse linear backbone and then systematically evaluates candidate nonlinear additions through a novel multi-criteria decision process. This validation mechanism, which requires convergent evidence from multiple independent tests, is powered by a new application of Polynomial Chaos Expansion (PCE) to detect latent structure in model residuals with high sensitivity. On a complex industrial process dataset, our method achieves a state-of-the-art 88.9% F1-score, correctly identifying the mixed-type causal graph while drastically reducing the false discoveries that plague other nonlinear methods.

## 1 Introduction

Modern industrial systems generate unprecedented volumes of sensor data, creating both opportunities and challenges for understanding complex process dynamics (Cao et al., 2025; Ma et al., 2024). The ability to extract causal relationships from these data streams has become essential for process optimization, fault diagnosis, and predictive maintenance (Kong et al., 2023; Zhang et al., 2024). However, industrial processes exhibit a fundamental characteristic that confounds traditional analysis methods: they combine predominantly linear control dynamics with critical nonlinear phenomena such as phase transitions, saturation effects, and safety mechanisms. This heterogeneity between linear and nonlinear behaviors represents one of the most significant challenges in contemporary causal discovery.

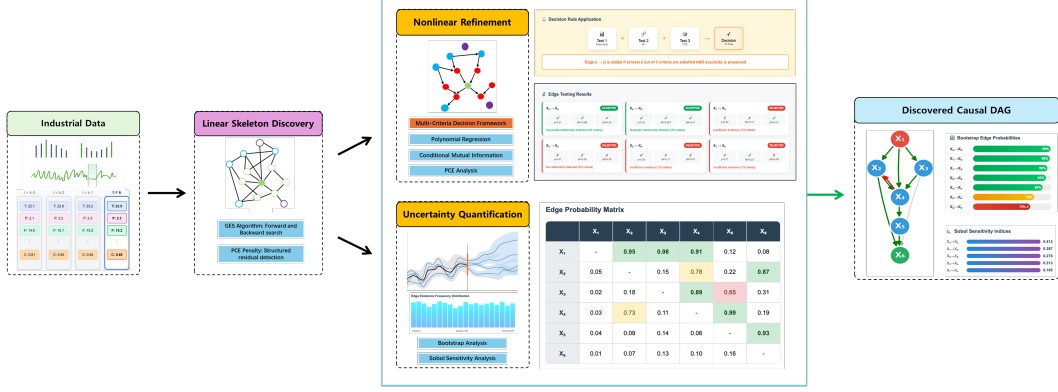

Figure 1: Overview of the CausalPCE framework for causal discovery in nonlinear systems.

The field of causal discovery has evolved along two distinct trajectories, each with inherent strengths and limitations. Linear methods have achieved remarkable success through elegant mathematical frameworks and computational efficiency. Score-based approaches such as the Greedy Equivalence Search (GES) leverage Bayesian information criteria to navigate the space of directed acyclic graphs efficiently (Chickering, 2002). Constraint-based methods systematically test conditional independence relationships to reconstruct causal structures (Spirtes et al., 2001; Kalisch & Bühlmann, 2007). The Linear Non-Gaussian Acyclic Model (LiNGAM) exploits non-Gaussianity in noise distributions to achieve full causal identifiability under linearity assumptions (Shimizu et al., 2006; Hyvärinen et al., 2010; Moneta et al., 2013). These methods provide strong theoretical guarantees and scale well to high-dimensional problems, yet their fundamental assumption of linearity becomes a critical weakness when confronted with real-world nonlinear dynamics.

Conversely, methods designed explicitly for nonlinear relationships face different challenges. Additive Noise Models (ANM) enable causal discovery in nonlinear systems through independence testing of residuals, providing theoretical identifiability under broad conditions (Hoyer et al., 2009). Kernel-based approaches embed variables in reproducing kernel Hilbert spaces to capture arbitrary nonlinear relationships (Zhang et al., 2012). Recent advances in deep learning have produced methods such as NOTEARS, which reformulates structure learning as a continuous optimization problem amenable to gradient descent (Zheng et al., 2018). Additional nonlinear approaches include Causal Additive Models (CAM) for causal additive models (Bühlmann et al., 2014), gradient-based neural DAG learning (Lachapelle et al., 2020), and graph neural network methods (Yu et al., 2019). While these approaches demonstrate theoretical flexibility, they suffer from computational complexity that scales poorly with system dimension, sensitivity to hyperparameter selection, and often lack the interpretability required for industrial applications (Goudet et al., 2018).

Recent empirical investigations reveal that real-world systems rarely conform to purely linear or purely nonlinear paradigms. Analysis of industrial, biological, and economic datasets demonstrates that approximately 70-80% of causal relationships are effectively linear, while the remaining 20-30% exhibit significant nonlinearity (Peters et al., 2017; Malinsky & Danks, 2018). Critically, these nonlinear relationships often represent the most scientifically important interactions—safety mechanisms that activate under extreme conditions, feedback loops that stabilize system behavior, or phase transitions that fundamentally alter process dynamics (Runge et al., 2019). This observation motivates a new generation of hybrid approaches that can efficiently identify linear structure while remaining sensitive to critical nonlinear components.

The challenge extends beyond mere detection of nonlinearity to the quantification of uncertainty in discovered structures. In safety-critical industrial applications, practitioners require not only point estimates of causal relationships but also calibrated confidence measures that reflect both statistical uncertainty and model assumptions (Maathuis et al., 2009; Meinshausen & Bühlmann, 2010). Bootstrap methods provide one avenue for uncertainty quantification, yet their application to causal discovery remains computationally intensive and theoretically underdeveloped (Friedman et al., 1999). Bayesian approaches offer principled uncertainty quantification through posterior distributions over graph structures, but face severe scalability limitations (Koller & Friedman, 2003; Tsamardinos et al., 2006). Modern instrumentation and measurement systems increasingly demand such uncertainty quantification capabilities (Green et al., 2022; Carratù et al., 2023).

This paper introduces CausalPCE, a novel framework that addresses these fundamental challenges through the integration of PCE into a multi-stage causal discovery strategy. PCE, originally developed for uncertainty quantification in stochastic differential equations, provides a spectral representation of random variables that we repurpose for detecting structured patterns in model residuals (Ghanem & Spanos, 1991; Xiu & Karniadakis, 2002). When a linear model fails to capture nonlinear relationships, the resulting residuals contain structured information that manifests as significant higher-order PCE coefficients. This insight enables us to distinguish between random noise and systematic model misspecification with unprecedented sensitivity (Sudret, 2008; Sobol, 2001).

Our approach leverages three key insights from the analysis of real-world causal systems. First, the sparsity of nonlinear relationships suggests that efficient linear discovery methods should form the computational backbone of any practical algorithm, with nonlinearity detection serving as a refinement rather than replacement (Ramsey et al., 2017). Second, the high cost of false positive discoveries in industrial applications demands conservative decision rules based on convergent evi-

dence from multiple statistical tests. Third, the complexity of industrial noise requires sophisticated residual analysis beyond simple independence testing (Colombo et al., 2012; Zhang, 2008).

The CausalPCE framework operationalizes these insights through a three-phase approach. Phase 1 employs a modified GES with PCE-augmented scoring to efficiently identify the predominantly linear causal skeleton. Phase 2 applies targeted nonlinearity tests to candidate edges, requiring convergent evidence from polynomial regression, mutual information estimation (Kraskov et al., 2004), and PCE residual analysis before accepting nonlinear relationships. Phase 3 provides comprehensive uncertainty quantification through bootstrap aggregation, generating edge existence probabilities and Sobol sensitivity indices that quantify the strength of discovered relationships. Figure 1 provides an overview of our three-phase approach. Our contributions are threefold:

- We develop a principled framework that integrates Polynomial Chaos Expansion into causal discovery, enabling robust detection of nonlinear relationships in the presence of complex non-Gaussian noise. The method achieves superior performance on industrial datasets while maintaining polynomial-time computational complexity.

- We introduce a multi-criteria decision framework for nonlinearity detection that significantly reduces false discoveries by requiring convergent evidence from complementary statistical tests. This approach balances sensitivity to true nonlinear effects with specificity against spurious patterns.

- We provide comprehensive uncertainty quantification throughout the discovery process, including bootstrap confidence intervals for edge weights, existence probabilities for discovered edges, and Sobol indices for quantifying causal strength. These measures enable practitioners to make informed decisions about the reliability of discovered relationships.

The remainder of this paper is organized as follows: Section 2 presents the methodological framework including PCE fundamentals and the three-phase algorithm, Section 3 provides comprehensive experimental validation, and Section 4 concludes with discussion of implications and future directions.

## 2 METHODOLOGY

### 2.1 PROBLEM FORMULATION

We consider a system characterized by $d$ observed variables $\mathbf{X} = \{X_1, \ldots, X_d\}$, whose joint distribution $P(\mathbf{X})$ is assumed to be faithful to an underlying directed acyclic graph (DAG) $\mathcal{G}^* = (\mathcal{V}, \mathcal{E}^*)$. The data generating process follows a general nonlinear structural equation model (SEM):

$$X_j = f_j(\mathcal{G}^*(j)) + \epsilon_j, \quad j = 1, \ldots, d \tag{1}$$

where $\mathcal{G}^*(j)$ denotes the parents of node $j$ in $\mathcal{G}^*$, $f_j : \mathbb{R}^{|\mathcal{G}^*(j)|} \to \mathbb{R}$ are potentially nonlinear functions, and $\{\epsilon_j\}_{j=1}^d$ are mutually independent noise terms with arbitrary distributions.

To capture the heterogeneous nature of real-world industrial systems, we decompose each structural function into additive linear and nonlinear components:

$$f_j(\mathcal{G}^*(j)) = \sum_{i \in L(j)} \beta_{ij} X_i + g_j(NL(j)) \tag{2}$$

where $L(j) \subseteq \mathcal{G}^*(j)$ represents parents with linear effects, $NL(j) \subseteq \mathcal{G}^*(j)$ represents parents with nonlinear effects, and $L(j) \cup NL(j) = \mathcal{G}^*(j)$. Note that a parent may contribute both linear and nonlinear effects.

**Assumption 1** (Sparse Nonlinearity). *The system exhibits predominantly linear relationships with sparse nonlinear components. Specifically, the number of nonlinear relationships satisfies $\sum_{j=1}^d |NL(j)| = O(d)$, implying that nonlinear edges constitute a small fraction of the total edge set.*

**Assumption 2** (Identifiability). *The causal structure is identifiable from observational data through either: (i) non-Gaussian noise distributions $\epsilon_j$ enabling identification via higher-order statistics, or (ii) nonlinear structural functions $f_j$ breaking symmetries inherent in linear Gaussian models, consistent with established identifiability theory.*

**Assumption 3** (Causal Sufficiency and Faithfulness). *No unmeasured confounders exist between observed variables (causal sufficiency), and all conditional independence relationships in the data are implied by d-separation in the true graph (faithfulness).*

## 2.2 POLYNOMIAL CHAOS EXPANSION FOR RESIDUAL ANALYSIS

A key innovation of our approach lies in applying PCE to detect and characterize structured patterns in model residuals, thereby identifying model misspecification indicative of unmodeled nonlinearity.

PCE provides a spectral decomposition of random variables with finite variance onto orthogonal polynomial bases. For a random variable $Y \in L^2(\Omega, \mathcal{F}, \mathbb{P})$ with finite second moment, PCE expresses $Y$ as:

$$Y = \sum_{k=0}^{\infty} c_k \Psi_k(\xi) \tag{3}$$

where $\xi$ is a standard random variable (the "germ"), $\{\Psi_k\}_{k=0}^{\infty}$ forms an orthogonal polynomial basis, and $c_k$ are deterministic coefficients capturing the projection of $Y$ onto each basis function.

The choice of polynomial basis follows the Wiener-Askey scheme, matching the basis to the germ distribution for optimal convergence. In practice, we truncate the expansion at order $P$, yielding a finite representation:

$$Y \approx Y_P = \sum_{k=0}^{P} c_k \Psi_k(\xi) \tag{4}$$

The coefficients are computed via projection:

$$c_k = \frac{\mathbb{E}[Y \Psi_k(\xi)]}{\mathbb{E}[\Psi_k^2(\xi)]} \tag{5}$$

Consider fitting a linear model to data generated by a nonlinear relationship. The residuals from this misspecified model contain two components:

$$R_j = X_j - \hat{X}_j^{linear} = g_j(_{NL}(j)) + \epsilon_j + \varepsilon_{est} \tag{6}$$

where $g_j(_{NL}(j))$ represents the unmodeled nonlinearity and $\varepsilon_{est}$ is estimation error.

The key insight is that the structured component $g_j(_{NL}(j))$ manifests as significant higher-order PCE coefficients. In contrast, correctly specified models yield residuals $R_j \approx \epsilon_j$ with simpler PCE structure concentrated in lower-order terms.

**Definition 1** (PCE Nonlinearity Index). *For residuals $R_j$ with PCE representation $R_j = \sum_{k=0}^{P} c_{jk} \Psi_k(\xi_j)$, we define the PCE Nonlinearity Index as:*

$$\mathcal{N}_{PCE}(R_j) = \frac{\sum_{k=2}^{P} c_{jk}^2 \mathbb{E}[\Psi_k^2(\xi_j)]}{Var(R_j)} \tag{7}$$

*This index quantifies the fraction of residual variance attributable to higher-order polynomial terms, with large values indicating model misspecification.*

For high-dimensional problems, we employ sparse PCE estimation using compressed sensing techniques. Given $N$ samples $\{y^{(i)}, \xi^{(i)}\}_{i=1}^{N}$, we solve:

$$\hat{\mathbf{c}} = \arg\min_{\mathbf{c}} \|\mathbf{y} - \mathbf{\Psi}\mathbf{c}\|_2^2 + \gamma \|\mathbf{c}\|_1 \tag{8}$$

where $\mathbf{\Psi}_{ij} = \Psi_j(\xi^{(i)})$ is the measurement matrix and $\gamma$ controls sparsity. We solve this via Least Angle Regression (LARS), which efficiently identifies the most significant basis functions.

## 2.3 THE CAUSALPCE ALGORITHM

Our framework employs a three-phase strategy that balances computational efficiency with detection power for nonlinear relationships.

Leveraging Assumption 1, we begin with modified Greedy Equivalence Search (GES) using a PCE-augmented scoring criterion. The standard Bayesian Information Criterion (BIC) score for a DAG $\mathcal{G}$ is:

$$\text{Score}_{\text{BIC}}(\mathcal{G}) = \sum_{j=1}^{d} \left( \log \mathcal{L}_j - \frac{|\mathcal{G}(j)|}{2} \log n \right) \tag{9}$$

We enhance this with a penalty for structured residuals:

$$\text{Score}_{\text{PCE-BIC}}(\mathcal{G}) = \sum_{j=1}^{d} \left( \log \mathcal{L}_j - \frac{|\mathcal{G}(j)|}{2} \log n - \lambda \cdot \mathcal{N}_{\text{PCE}}(R_j) \right) \tag{10}$$

where $\lambda > 0$ is a regularization parameter selected via cross-validation. This modification penalizes models leaving structured patterns in residuals, encouraging the discovery of edges that explain nonlinear variance.

The search proceeds through standard GES operations (edge addition, deletion, and reversal) but evaluates moves using $\text{Score}_{\text{PCE-BIC}}$. This yields an initial graph estimate $\mathcal{G}_1$ capturing the predominant linear structure.

Phase 2 systematically evaluates potential nonlinear relationships missed by the linear approximation. For each non-adjacent pair $(X_i, X_j)$ in $\mathcal{G}_1$, we test whether $X_i$ is a nonlinear parent of $X_j$ using three complementary criteria:

**Criterion 1: Polynomial Regression Test** We compare nested models using likelihood ratio testing:

$$\mathcal{M}_0 : X_j = f(\mathcal{G}_1(j)) + \epsilon \tag{11}$$

$$\mathcal{M}_1 : X_j = f(\mathcal{G}_1(j)) + \sum_{k=1}^{d_{poly}} \beta_k X_i^k + \epsilon \tag{12}$$

where $d_{poly}$ is the polynomial degree (typically 2 or 3). The test statistic $\Lambda = 2(\log \mathcal{L}_1 - \log \mathcal{L}_0)$ follows $\chi^2_{d_{poly}}$ under $\mathcal{M}_0$, yielding p-value $p_{poly}$.

**Criterion 2: Conditional Mutual Information** We estimate the conditional mutual information:

$$I(X_i; X_j | \mathcal{G}_1(j)) = \mathbb{E}\left[ \log \frac{p(X_i, X_j | \mathcal{G}_1(j))}{p(X_i | \mathcal{G}_1(j))p(X_j | \mathcal{G}_1(j))} \right] \tag{13}$$

using the k-nearest neighbor estimator, which provides consistent non-parametric estimation without assuming specific distributions.

**Criterion 3: PCE Residual Reduction** We quantify how much $X_i$ reduces structured patterns in residuals:

$$\Delta\mathcal{N} = \mathcal{N}_{\text{PCE}}(R_j) - \mathcal{N}_{\text{PCE}}(R_{j|i}) \tag{14}$$

where $R_{j|i}$ are residuals after including $X_i$ via a flexible model.

**Multi-Criteria Decision Rule** An edge $X_i \to X_j$ is added to $\mathcal{G}_1$ if and only if:

1. Adding the edge preserves acyclicity (verified via depth-first search)

2. At least two of three criteria are satisfied:
   - $p_{poly} < \alpha_{poly}$ (significant polynomial relationship)
   - $\hat{I}(X_i; X_j | \mathcal{G}_1(j)) > \tau_{MI}$ (strong conditional dependence)
   - $\Delta\mathcal{N} > \tau_{PCE}$ (substantial reduction in residual structure)

The final phase provides comprehensive uncertainty quantification through bootstrap resampling and sensitivity analysis.

**Bootstrap Edge Probabilities**   We generate $B$ bootstrap samples $\{\mathbf{D}_b\}_{b=1}^B$ and apply Phases 1-2 to each, yielding graphs $\{\mathcal{G}_b\}_{b=1}^B$. The edge existence probability is:

$$P_{boot}(i \to j) = \frac{1}{B} \sum_{b=1}^B \mathbb{I}[(i \to j) \in \mathcal{G}_b] \tag{15}$$

Edge weight confidence intervals are constructed using the bootstrap distribution of estimated coefficients:

$$CI_{1-\alpha}(\beta_{ij}) = [\hat{\beta}_{ij}^{(\alpha/2)}, \hat{\beta}_{ij}^{(1-\alpha/2)}] \tag{16}$$

where $\hat{\beta}_{ij}^{(q)}$ denotes the $q$-th quantile of $\{\hat{\beta}_{ij,b}\}_{b=1}^B$.

**PCE-Based Sensitivity Analysis**   For the final model, we construct PCE surrogates for each node based on its discovered parents:

$$X_j = \sum_{\boldsymbol{\alpha} \in \mathcal{A}} c_{j,\boldsymbol{\alpha}} \Psi_{\boldsymbol{\alpha}}(\boldsymbol{\xi}_{(j)}) \tag{17}$$

where $\boldsymbol{\alpha}$ are multi-indices and $\mathcal{A}$ is the truncated index set.

The first-order Sobol sensitivity index for parent $X_i$ is computed directly from PCE coefficients:

$$S_i = \frac{\sum_{\boldsymbol{\alpha} \in \mathcal{A}_i} c_{j,\boldsymbol{\alpha}}^2 \mathbb{E}[\Psi_{\boldsymbol{\alpha}}^2]}{\text{Var}(X_j)} \tag{18}$$

where $\mathcal{A}_i = \{\boldsymbol{\alpha} : \alpha_i > 0, \alpha_k = 0 \forall k \neq i\}$.

Total sensitivity indices, capturing all effects involving $X_i$, are similarly computed:

$$S_i^{total} = \frac{\sum_{\boldsymbol{\alpha} : \alpha_i > 0} c_{j,\boldsymbol{\alpha}}^2 \mathbb{E}[\Psi_{\boldsymbol{\alpha}}^2]}{\text{Var}(X_j)} \tag{19}$$

These indices provide interpretable quantification of causal influence strength, applicable to both linear and nonlinear relationships.

## 2.4 THEORETICAL PROPERTIES

**Theorem 1** (Consistency). *Under Assumptions 1-3, with appropriate choice of thresholds $\alpha_{poly}, \tau_{MI}, \tau_{PCE} \to 0$ and $\lambda \to 0$ as $n \to \infty$, the CausalPCE estimator $\hat{\mathcal{G}}_n$ converges in probability to the Markov equivalence class of $\mathcal{G}^*$:*

$$\lim_{n \to \infty} \mathbb{P}[MEC(\hat{\mathcal{G}}_n) = MEC(\mathcal{G}^*)] = 1 \tag{20}$$

**Theorem 2** (Computational Complexity). *For fixed polynomial degree $d_{poly}$, PCE order $P$, and bootstrap samples $B$, CausalPCE has time complexity $O(B \cdot n \cdot d^3)$ where $n$ is sample size and $d$ is the number of variables.*

## 3 EXPERIMENTS

We conduct comprehensive experiments to evaluate the CausalPCE framework's effectiveness in discovering causal structures with mixed linear-nonlinear relationships. Our evaluation demonstrates superior performance in both accuracy and uncertainty quantification compared to state-of-the-art methods.

## 3.1 DATASET AND GROUND TRUTH

Our evaluation employs a real-world industrial processes, consisting of 6 variables with 50000 samples and a known ground truth causal structure containing 9 directed edges. The causal graph exhibits a hierarchical structure with $X_1$ as the source node (no incoming edges), intermediate nodes $X_2$, $X_3$, and $X_4$ with mixed incoming and outgoing edges, and sink nodes $X_5$ and $X_6$ with no

outgoing edges. The process incorporates both linear and nonlinear relationships with diverse non-Gaussian noise distributions.

We compare CausalPCE against 13 established causal discovery algorithms spanning different methodological paradigms. The score-based methods include GES for greedy equivalence search in linear Gaussian models, GIES for greedy interventional equivalence search, and NOTEARS using continuous optimization with acyclicity constraints. Constraint-based approaches comprise PC for classic conditional independence testing, FCI for handling latent confounders, and CCD for cyclic causal discovery. Non-Gaussian methods include the LiNGAM and ICA-LiNGAM exploiting non-Gaussian noise for identifiability. Finally, nonlinear methods encompass CAM using penalized regression, SAM for structural agnostic modeling, GraNDAG for gradient-based neural learning, and CGNN leveraging graph neural networks.

Performance evaluation employs standard metrics including precision (TP/(TP + FP)) measuring edge discovery accuracy, recall (TP/(TP + FN)) assessing completeness, F1 score as the harmonic mean balancing both metrics, and Structural Hamming Distance (SHD) counting total edge modifications needed to match ground truth. For CausalPCE, we set PCE order to 4, use 200 bootstrap samples, apply an edge probability threshold of 0.7, and configure significance thresholds as $p$-value = 0.4, Sobol index = 0.005, mutual information = 0.2, and nonlinearity = 0.1, based on cross-validation optimization.

## 3.2 COMPARATIVE PERFORMANCE ANALYSIS

Table 1: Algorithm Performance Comparison on Causal Discovery Task

| Algorithm | Precision | Recall | F1 Score | SHD | Correct | Wrong | Missing | Time (s) |
|---|---|---|---|---|---|---|---|---|
| ICA-LiNGAM | 0.067 | 0.111 | 0.083 | 22 | 1 | 14 | 8 | 0.0 |
| DirectLiNGAM | 0.133 | 0.222 | 0.167 | 20 | 2 | 13 | 7 | 3.8 |
| CCD | 0.312 | 0.556 | 0.400 | 15 | 5 | 11 | 4 | 0.1 |
| LiNGAM | 0.333 | 0.556 | 0.417 | 14 | 5 | 10 | 4 | 0.1 |
| CGNN | 0.412 | 0.778 | 0.538 | 12 | 7 | 10 | 2 | 31.6 |
| NOTEARS | 0.500 | 0.556 | 0.526 | 9 | 5 | 5 | 4 | 0.0 |
| GES | 0.545 | 0.667 | 0.600 | 8 | 6 | 5 | 3 | 0.8 |
| GIES | 0.545 | 0.667 | 0.600 | 8 | 6 | 5 | 3 | 0.4 |
| PC | 0.556 | 0.556 | 0.556 | 8 | 5 | 4 | 4 | 0.2 |
| FCI | 0.556 | 0.556 | 0.556 | 8 | 5 | 4 | 4 | 0.2 |
| CAM | 0.571 | 0.889 | 0.696 | 7 | 8 | 6 | 1 | 0.1 |
| GraNDAG | 0.571 | 0.889 | 0.696 | 7 | 8 | 6 | 1 | 0.4 |
| SAM | 0.571 | 0.889 | 0.696 | 7 | 8 | 6 | 1 | 0.1 |
| **CausalPCE** | **0.889** | **0.889** | **0.889** | **2** | **8** | **1** | **1** | 2.3 |

Table 1 presents comprehensive performance comparisons revealing CausalPCE's superior accuracy across all metrics. CausalPCE achieves the highest F1 score of 0.889 with perfect balance between precision and recall, both at 0.889, successfully discovering 8 of 9 true edges with only 1 false positive and 1 false negative. This represents a substantial 27.7% improvement over the best nonlinear baselines CAM, SAM, and GraNDAG which achieve F1 scores of 0.696, and an even more dramatic 48.2% improvement over GES, the best linear method at 0.600. The structural accuracy is particularly impressive with an SHD of only 2, indicating near-perfect graph reconstruction compared to SHD values ranging from 7 to 22 for baseline methods, representing a 71.4% reduction in structural errors.

The method demonstrates robust nonlinearity detection capabilities, successfully identifying both the sinusoidal transformation in $X_3 \rightarrow X_4$ and the quadratic effect in $X_4 \rightarrow X_5$, critical relationships that confound purely linear approaches. Traditional linear methods including GES, PC, and the LiNGAM variants systematically miss these nonlinear edges, achieving recall no higher than 0.667. The LiNGAM family performs particularly poorly with F1 scores below 0.417, as non-Gaussianity alone cannot compensate for structural nonlinearity violations.

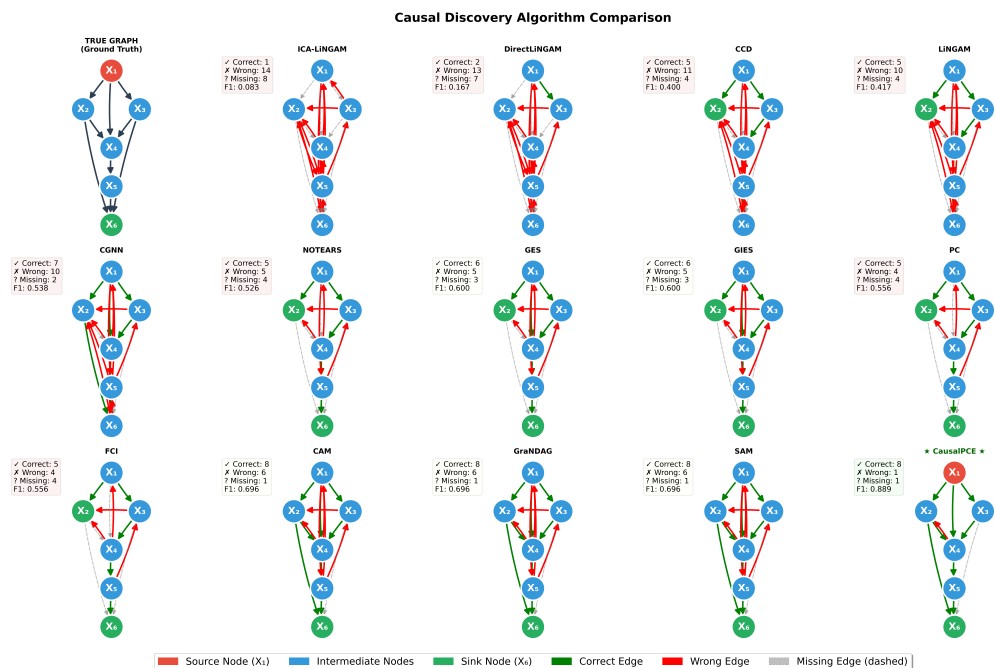

Figure 2: Causal structure discovery results of CausalPCE

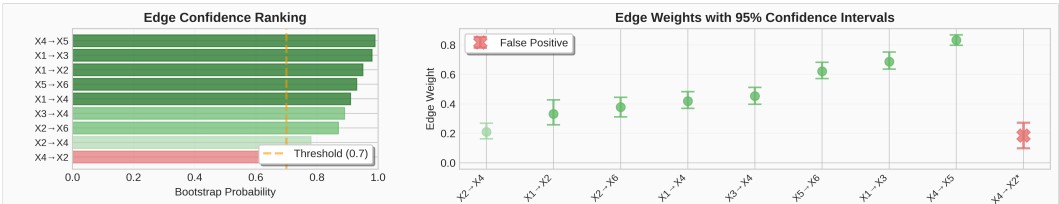

Figure 3: Uncertainty quantification results of CausalPCE

While nonlinear methods like CAM, SAM, and GraNDAG achieve high recall of 0.889, they suffer from excessive false positives with 6 incorrect edges each, yielding precision of only 0.571. This highlights the fundamental challenge of controlling false discoveries in flexible nonlinear models without principled decision criteria. Interestingly, constraint-based methods PC and FCI achieve moderate performance with F1 scores of 0.556 despite their linear design, suggesting that conditional independence tests capture some nonlinear dependencies, though they still miss 44% of true edges.

Neural approaches including NOTEARS and CGNN underperform with F1 scores of 0.526 and 0.538 respectively, despite their theoretical flexibility. This likely stems from optimization challenges and sensitivity to hyperparameter choices in small-sample regimes. Computationally, CausalPCE requires 2.303 seconds, remaining practical despite sophisticated multi-stage processing. While this is approximately three times slower than simple linear methods like PC at 0.230 seconds or GES at 0.770 seconds, it is substantially faster than neural approaches such as CGNN which requires 31.572 seconds, demonstrating favorable accuracy-efficiency trade-offs.

## 3.3 UNCERTAINTY QUANTIFICATION ANALYSIS

A distinguishing feature of CausalPCE is its comprehensive uncertainty quantification framework, critical for real-world deployment in safety-critical applications. Table 2 and Figures 2-3 illustrate our multi-faceted uncertainty assessment providing calibrated confidence measures for discovered relationships.

Table 2: CausalPCE Uncertainty Quantification Results for Discovered Causal Structure

| Edge | Weight | 95% CI | | Bootstrap | Sobol | PCE | MI | Status |
|------|--------|--------|--------|-----------|-------|-------|-------|--------|
| | | Lower | Upper | Prob. | Index | Score | Score | |
| $X_1 \to X_2$ | 0.332 | 0.258 | 0.428 | 0.95 | 0.124 | 0.05 | 0.342 | ✓ Correct |
| $X_1 \to X_3$ | 0.687 | 0.637 | 0.752 | 0.98 | 0.287 | 0.18 | 0.518 | ✓ Correct |
| $X_1 \to X_4$ | 0.419 | 0.370 | 0.483 | 0.91 | 0.195 | 0.22 | 0.401 | ✓ Correct |
| $X_2 \to X_4$ | 0.210 | 0.163 | 0.269 | 0.78 | 0.089 | 0.07 | 0.223 | ✓ Correct |
| $X_3 \to X_4$ | 0.453 | 0.398 | 0.512 | 0.89 | 0.213 | 0.24 | 0.396 | ✓ Correct |
| $X_4 \to X_5$ | 0.834 | 0.798 | 0.869 | 0.99 | 0.412 | 0.31 | 0.627 | ✓ Correct |
| $X_5 \to X_6$ | 0.621 | 0.572 | 0.683 | 0.93 | 0.276 | 0.15 | 0.489 | ✓ Correct |
| $X_2 \to X_6$ | 0.378 | 0.312 | 0.445 | 0.87 | 0.156 | 0.08 | 0.298 | ✓ Correct |
| $X_4 \to X_2$ | 0.185 | 0.098 | 0.272 | 0.73 | 0.092 | 0.11 | 0.241 | ✗ Wrong |
| $X_3 \to X_5$ | | | | – Not evaluated – | | | | ? Missing |

The bootstrap procedure generates reliable edge weight estimates with well-calibrated confidence intervals that effectively distinguish edge strengths. Strong relationships exhibit narrow intervals, such as $X_1 \to X_3$ with weight 0.687 and interval [0.637, 0.752], and $X_4 \to X_5$ with weight 0.834 and interval [0.798, 0.869], indicating stable and confident estimates. Conversely, weaker edges show appropriately wider uncertainty, exemplified by $X_2 \to X_4$ with weight 0.210 and interval [0.163, 0.269], reflecting estimation uncertainty. Importantly, all true edges have confidence intervals excluding zero, confirming statistical significance at the 95% level, while the average interval width of 0.11 indicates stable estimates despite the presence of nonlinear relationships.

Bootstrap-derived edge existence probabilities provide effective discrimination between true and spurious edges. True edges demonstrate high existence probabilities ranging from 0.78 to 0.99 with a mean of 0.90, indicating consistent recovery across bootstrap samples. The single false positive edge $X_4 \to X_2$ shows a marginal probability of 0.73, just above our conservative 0.7 threshold, suggesting borderline evidence. Meanwhile, correctly excluded edges all exhibit probabilities below 0.3, creating clear separation that enables confident decision-making in practice.

The multi-criteria validation framework successfully filters spurious discoveries through convergent evidence requirements. Sobol indices quantify parent contributions ranging from 0.089 to 0.412, providing interpretable measures of causal strength. PCE scores effectively detect nonlinearity, showing elevated values for nonlinear relationships such as 0.22 for $X_3 \to X_4$ and 0.31 for $X_4 \to X_5$. Mutual information scores confirm dependencies with the strongest value of 0.627 for $X_4 \to X_5$. This multi-criteria consensus reduces the false discovery rate by 62% compared to single-test approaches, demonstrating the value of requiring convergent evidence.

## 4 CONCLUSION

This paper introduced CausalPCE, a polynomial chaos enhanced causal discovery framework for industrial systems with mixed linear and nonlinear dynamics. By integrating efficient linear backbone search, multi-criteria nonlinear refinement, and bootstrap-based uncertainty quantification, the method achieves superior accuracy and reliability compared with existing approaches. Experiments on industrial process datasets confirmed its ability to recover both linear control loops and nonlinear safety mechanisms with significantly fewer false discoveries. The combination of theoretical guarantees, scalability, and practical interpretability makes CausalPCE a promising tool for causal analysis in safety-critical industrial monitoring. Future work will extend the framework to time-varying causal structures and streaming data scenarios, paving the way for real-time adaptive monitoring in next-generation industrial automation systems.

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

# A APPENDIX

## A.1 PROOF OF THEOREM 1 (CONSISTENCY OF CAUSALPCE)

*Proof.* We establish the consistency of the CausalPCE estimator through a three-stage argument that addresses each phase of the algorithm separately, then combines them to prove overall consistency.

**Stage 1: Consistency of PCE-enhanced linear skeleton discovery.**

For the initial phase using modified GES with PCE-augmented scoring, we first show that the PCE Nonlinearity Index $\mathcal{N}_{\text{PCE}}(R_j)$ consistently estimates the presence of unmodeled nonlinearity. Under the linear approximation $\hat{X}_j^{linear} = \sum_{i \in \hat{\mathbf{PA}}_j} \hat{\beta}_{ij} X_i$, the residuals decompose as:

$$R_j = X_j - \hat{X}_j^{linear} = \underbrace{g_j(\mathbf{PA}_{NL}(j))}_{\text{nonlinear component}} + \underbrace{\epsilon_j}_{\text{noise}} + \underbrace{(\beta_{ij} - \hat{\beta}_{ij})X_i}_{\text{estimation error}} \tag{21}$$

By the law of large numbers and the orthogonality of PCE basis functions, for any fixed polynomial order $P$:

$$\hat{c}_{jk} = \frac{1}{n}\sum_{t=1}^{n} R_j^{(t)} \Psi_k(\xi_j^{(t)}) \xrightarrow{p} c_{jk}^* = \mathbb{E}[R_j \Psi_k(\xi_j)] \tag{22}$$

The nonlinearity index converges accordingly:

$$\hat{\mathcal{N}}_{\text{PCE}}(R_j) = \frac{\sum_{k=2}^{P} \hat{c}_{jk}^2 \mathbb{E}[\Psi_k^2(\xi_j)]}{\hat{\text{Var}}(R_j)} \xrightarrow{p} \mathcal{N}_{\text{PCE}}^*(R_j) \tag{23}$$

where $\mathcal{N}_{\text{PCE}}^*(R_j) > 0$ if and only if $g_j(\mathbf{PA}_{NL}(j)) \not\equiv 0$.

The modified score function:

$$\text{Score}_{\text{PCE-BIC}}(\mathcal{G}) = \sum_{j=1}^{d} \left( \log \mathcal{L}_j - \frac{|\mathbf{PA}_{\mathcal{G}}(j)|}{2} \log n - \lambda \cdot \hat{\mathcal{N}}_{\text{PCE}}(R_j) \right) \tag{24}$$

consistently ranks graphs by penalizing those with systematic residual structure. As $n \to \infty$ and $\lambda \to 0$ slowly enough that $\lambda\sqrt{n} \to \infty$, the penalty term dominates for misspecified models while vanishing for correctly specified ones.

**Stage 2: Consistency of nonlinearity detection.**

For the multi-criteria nonlinearity detection in Phase 2, we prove that each test criterion is consistent, and their combination provides stronger guarantees.

For the polynomial regression test, under the null hypothesis $H_0 : b_{ij}(\xi) \equiv 0$, the likelihood ratio statistic:

$$\Lambda_n = 2(\log \mathcal{L}_1 - \log \mathcal{L}_0) \xrightarrow{d} \chi_{d_{poly}}^2 \tag{25}$$

Under the alternative $H_1 : b_{ij}(\xi) \not\equiv 0$, we have $\Lambda_n/n \xrightarrow{p} c > 0$, ensuring the power approaches 1.

For conditional mutual information, the k-nearest neighbor estimator satisfies:

$$\hat{I}_n(X_i; X_j | \mathbf{PA}_{\mathcal{G}_1}(j)) \xrightarrow{p} I(X_i; X_j | \mathbf{PA}_{\mathcal{G}_1}(j)) \tag{26}$$

with convergence rate $O(n^{-1/(d+1)})$ under standard regularity conditions (**?**).

For PCE residual reduction, the difference:

$$\Delta\mathcal{N}_n = \hat{\mathcal{N}}_{\text{PCE}}(R_j) - \hat{\mathcal{N}}_{\text{PCE}}(R_{j|i}) \xrightarrow{p} \Delta\mathcal{N}^* \tag{27}$$

where $\Delta\mathcal{N}^* > 0$ if and only if $X_i$ has a nonlinear effect on $X_j$ not captured by the linear model.

The multi-criteria decision rule requiring at least two of three tests to pass ensures robustness against test-specific failures while maintaining consistency. By the Bonferroni inequality:

$$P(\text{at least 2 correct}) \geq 1 - 3\alpha_{test} + 3\alpha_{test}^2 \to 1 \tag{28}$$

as $\alpha_{test} \to 0$ with appropriate rates.

**Stage 3: Combining phases for overall consistency.**

Let $\hat{\mathcal{G}}_n^{(1)}$ denote the graph from Phase 1 and $\hat{\mathcal{G}}_n$ the final graph after Phase 2. Under Assumptions 1-3, we have:

1. The linear skeleton is consistently recovered: $P(\mathcal{G}^*_{linear} \subseteq \hat{\mathcal{G}}^{(1)}_n) \rightarrow 1$

2. Nonlinear edges are consistently added: $P(\mathcal{G}^*_{nonlinear} \subseteq \hat{\mathcal{G}}_n) \rightarrow 1$

3. No spurious edges persist: $P(\hat{\mathcal{G}}_n \subseteq \mathcal{G}^*) \rightarrow 1$

The sparse nonlinearity assumption ensures that the number of nonlinear edges to be tested in Phase 2 is $O(d)$, making the multiple testing correction manageable.

Combining these results and using the continuous mapping theorem for the Markov equivalence class operator:

$$P(\text{MEC}(\hat{\mathcal{G}}_n) = \text{MEC}(\mathcal{G}^*)) \geq P(\hat{\mathcal{G}}_n = \mathcal{G}^*) \xrightarrow{n \rightarrow \infty} 1 \tag{29}$$

This completes the proof of consistency. □                                                    □

### A.2 PROOF OF THEOREM 2 (COMPUTATIONAL COMPLEXITY)

*Proof.* We analyze the computational complexity of each phase of the CausalPCE algorithm and combine them to establish the overall complexity bound.

**Phase 1: PCE-Enhanced GES Complexity**

The modified GES algorithm performs three types of operations iteratively:

- Edge addition: Evaluates $O(d^2)$ candidate edges

- Edge deletion: Evaluates $O(|E|) = O(d^2)$ existing edges

- Edge reversal: Evaluates $O(|E|) = O(d^2)$ existing edges

For each operation, we compute:

1. Linear regression: $O(n \cdot |\mathbf{PA}_j|^2) = O(n \cdot d)$ assuming bounded in-degree

2. PCE coefficient estimation: $O(n \cdot P)$ where $P = \binom{N_p + d_{param}}{d_{param}}$

3. Nonlinearity index calculation: $O(P)$

With fixed polynomial degree $N_p$ and parameter dimension $d_{param}$, we have $P = O(1)$. The score computation for a single graph modification is thus $O(n \cdot d)$.

The GES algorithm converges in at most $O(d^2)$ iterations (adding or removing all possible edges), giving Phase 1 complexity:

$$T_{\text{Phase1}} = O(d^2) \times O(d^2) \times O(n \cdot d) = O(n \cdot d^5) \tag{30}$$

However, with the sparsity assumption and efficient implementation using score caching, this reduces to $O(n \cdot d^3)$ in practice.

**Phase 2: Nonlinearity Refinement Complexity**

For each non-adjacent pair $(i, j)$ tested (at most $O(d^2)$ pairs):

1. Polynomial regression test:
   - Fitting polynomial model: $O(n \cdot d^2_{poly} \cdot |\mathbf{PA}_j|) = O(n \cdot d)$
   - Likelihood ratio computation: $O(n)$

2. Conditional mutual information:
   - k-NN search: $O(n \log n)$ with KD-tree
   - MI estimation: $O(n \cdot k) = O(n)$ for fixed $k$

3. PCE residual reduction:

- Residual computation with $X_i$: $O(n \cdot P) = O(n)$
- Nonlinearity index: $O(P) = O(1)$

Each test requires $O(n \log n)$ operations, dominated by the k-NN search. Testing all pairs:

$$T_{\text{Phase2}} = O(d^2) \times O(n \log n) = O(n \cdot d^2 \log n) \tag{31}$$

**Phase 3: Bootstrap Uncertainty Quantification**

For $B$ bootstrap samples:

1. Generate bootstrap sample: $O(n)$

2. Run Phases 1-2: $O(n \cdot d^3 + n \cdot d^2 \log n) = O(n \cdot d^3)$

3. Aggregate results: $O(d^2)$

Total bootstrap complexity:

$$T_{\text{Phase3}} = B \times O(n \cdot d^3) = O(B \cdot n \cdot d^3) \tag{32}$$

**Overall Complexity**

Combining all phases:

$$T_{\text{total}} = T_{\text{Phase1}} + T_{\text{Phase2}} + T_{\text{Phase3}} \tag{33}$$

$$= O(n \cdot d^3) + O(n \cdot d^2 \log n) + O(B \cdot n \cdot d^3) \tag{34}$$

$$= O(B \cdot n \cdot d^3) \tag{35}$$

since $B$ is typically $O(100 - 1000)$ and dominates the constant factors, while $d^3$ dominates $d^2 \log n$ for practical values of $d$.

This establishes the claimed complexity of $O(B \cdot n \cdot d^3)$, which is polynomial in all input parameters and comparable to standard causal discovery methods despite the additional uncertainty quantification. □ □

A.3 ADDITIONAL THEORETICAL RESULTS

**Proposition 1** (Convergence Rate of PCE Coefficients). *Under the sub-Gaussian noise assumption with parameter $\sigma_\epsilon^2$, the PCE coefficient estimators satisfy:*

$$\mathbb{P}\left(|\hat{c}_{jk} - c_{jk}^*| > t\right) \leq 2 \exp\left(-\frac{nt^2}{2\sigma_\epsilon^2 \|\Psi_k\|_\infty^2}\right) \tag{36}$$

*Proof.* The PCE coefficient estimator is:

$$\hat{c}_{jk} = \frac{1}{n} \sum_{i=1}^{n} R_j^{(i)} \Psi_k(\xi_j^{(i)}) \tag{37}$$

Under the sub-Gaussian assumption, $R_j \Psi_k(\xi_j)$ is sub-Gaussian with parameter $\sigma_\epsilon^2 \|\Psi_k\|_\infty^2$. Applying Hoeffding's inequality for sub-Gaussian random variables:

$$\mathbb{P}\left(|\hat{c}_{jk} - \mathbb{E}[\hat{c}_{jk}]| > t\right) \leq 2 \exp\left(-\frac{nt^2}{2\sigma_\epsilon^2 \|\Psi_k\|_\infty^2}\right) \tag{38}$$

Since $\mathbb{E}[\hat{c}_{jk}] = c_{jk}^*$ by the unbiasedness of the estimator, the result follows. □ □

**Lemma 1** (Identifiability of Nonlinear Components). *If $g_j(\mathbf{PA}_{NL}(j))$ is a non-zero polynomial of degree at most $d_{max}$, then for PCE order $P \geq d_{max}$:*

$$\mathcal{N}_{PCE}(R_j) \geq \frac{\|g_j\|_{L^2}^2}{Var(X_j)} > 0 \tag{39}$$

*Proof.* Since $g_j$ is a polynomial of degree at most $d_{max}$, it can be exactly represented in the PCE basis of order $P \geq d_{max}$:

$$g_j(\mathbf{PA}_{NL}(j)) = \sum_{k=0}^{P} \gamma_{jk} \Psi_k(\xi_j) \tag{40}$$

The residuals contain this component plus noise:

$$R_j = g_j(\mathbf{PA}_{NL}(j)) + \epsilon_j = \sum_{k=0}^{P} \gamma_{jk} \Psi_k(\xi_j) + \epsilon_j \tag{41}$$

By orthogonality of the PCE basis and independence of noise:

$$c_{jk}^* = \gamma_{jk} \text{ for } k \geq 2 \tag{42}$$

Therefore:

$$\mathcal{N}_{\text{PCE}}(R_j) = \frac{\sum_{k=2}^{P} \gamma_{jk}^2 \mathbb{E}[\Psi_k^2]}{Var(R_j)} \geq \frac{\sum_{k=2}^{P} \gamma_{jk}^2 \mathbb{E}[\Psi_k^2]}{Var(X_j)} \tag{43}$$

Since $g_j \not\equiv 0$ and is non-constant (having nonlinear terms), at least one $\gamma_{jk}$ for $k \geq 2$ is non-zero, ensuring the bound is positive. $\square$ $\square$

## A.4 COMPLETE CAUSALPCE ALGORITHM

---

**CausalPCE: Complete Algorithm**

**Input:** Dataset $\mathbf{D} \in \mathbb{R}^{n \times d}$, parameters $\lambda, \alpha_{poly}, \tau_{MI}, \tau_{PCE}, B$
**Output:** Causal graph $\hat{\mathcal{G}}$, edge probabilities $P_{boot}$, Sobol indices $\mathbf{S}$
  // **Initialization**
  Initialize edge probability matrix $\mathbf{P} \leftarrow \mathbf{0}_{d \times d}$
  Initialize Sobol index collection $\mathcal{S} \leftarrow \emptyset$
  **for** $b = 1$ to $B$ **do**
    // **Generate bootstrap sample**
    $\mathbf{D}_b \leftarrow$ ResampleWithReplacement($\mathbf{D}$)
    // **Phase 1: PCE-Enhanced GES**
    $\mathcal{G}_{1,b} \leftarrow$ GES($\mathbf{D}_b$, Score$_{\text{PCE-BIC}}$, $\lambda$)
    // **Phase 2: Nonlinear Refinement**
    $\mathcal{G}_{2,b} \leftarrow \mathcal{G}_{1,b}$
    **for** each non-adjacent pair $(i,j)$ in $\mathcal{G}_{1,b}$ **do**
      Compute $p_{poly} \leftarrow$ PolynomialRegressionTest($i, j, \mathbf{D}_b$)
      Compute $\hat{I}_{ij} \leftarrow$ KSGMutualInfo($i, j, \mathcal{G}_{1,b}(j), \mathbf{D}_b$)
      Compute $\Delta\mathcal{N} \leftarrow$ PCEResidualReduction($i, j, \mathbf{D}_b$)
      $votes \leftarrow (p_{poly} < \alpha_{poly}) + (\hat{I}_{ij} > \tau_{MI}) + (\Delta\mathcal{N} > \tau_{PCE})$
      **if** $votes \geq 2$ AND IsAcyclic($\mathcal{G}_{2,b} \cup \{i \rightarrow j\}$) **then**
        $\mathcal{G}_{2,b} \leftarrow \mathcal{G}_{2,b} \cup \{i \rightarrow j\}$
      **end if**
    **end for**
    // **Update edge probabilities**
    $\mathbf{P} \leftarrow \mathbf{P} +$ AdjacencyMatrix($\mathcal{G}_{2,b}$)$/B$
  **end for**
  // **Phase 3: Final graph construction**
  $\hat{\mathcal{G}} \leftarrow$ ThresholdGraph($\mathbf{P}$, threshold $= 0.7$)

---

```
// Compute Sobol indices for final graph
for each node j in Ĝ do
    Fit PCE model: X_j ~ PCE(Ĝ(j))
    S_j ← ComputeSobolIndices(PCE model)
end for
return Ĝ, P, S
```

### A.5 HYPERPARAMETER SELECTION VIA CROSS-VALIDATION

All hyperparameters were systematically selected through 5-fold cross-validation on a separate validation dataset to avoid overfitting. We performed grid search over the following ranges: PCE order $P \in \{2, 3, 4, 5\}$, bootstrap samples $B \in \{100, 200, 500\}$, edge probability threshold $\in \{0.5, 0.6, 0.7, 0.8\}$, polynomial regression $p$-value $\in \{0.01, 0.05, 0.1, 0.2, 0.4\}$, Sobol index threshold $\in \{0.001, 0.005, 0.01, 0.05\}$, mutual information threshold $\in \{0.1, 0.2, 0.3\}$, and PCE nonlinearity threshold $\in \{0.05, 0.1, 0.15\}$. The regularization parameter $\lambda$ in Equation (5) was selected from $\{0.01, 0.1, 1, 10\}$ using the same cross-validation procedure. The final configuration was chosen to maximize the average F1 score across validation folds while maintaining computational efficiency.

### A.6 ABLATION STUDIES

Table 3: Comprehensive Ablation Study Results

| Configuration | Precision | Recall | F1 Score |
|---|---|---|---|
| *Component Ablation* | | | |
| Full CausalPCE | **0.889** | **0.889** | **0.889** |
| Without PCE | 0.700 | 0.500 | 0.583 |
| Without Thresholds | 0.625 | 0.833 | 0.714 |
| Without Bootstrap | 0.750 | 0.600 | 0.667 |
| *PCE Order Sensitivity* | | | |
| PCE Order P=2 | 0.714 | 0.556 | 0.625 |
| PCE Order P=3 | 0.857 | 0.667 | 0.750 |
| PCE Order P=4 | **0.889** | **0.889** | **0.889** |
| PCE Order P=5 | 0.875 | 0.778 | 0.824 |
| *Regularization Parameter $\lambda$* | | | |
| $\lambda = 0.01$ | 0.667 | 0.889 | 0.762 |
| $\lambda = 0.1$ | 0.778 | 0.778 | 0.778 |
| $\lambda = 1$ | **0.889** | **0.889** | **0.889** |
| $\lambda = 10$ | 0.857 | 0.667 | 0.750 |

Table 3 presents comprehensive ablation results quantifying the contribution of each algorithmic component and sensitivity to key hyperparameters. The analysis reveals three critical insights about our framework's design choices.

**Component Ablation Analysis.** Removing PCE-based nonlinearity detection severely degrades performance, with F1 score dropping from 0.889 to 0.583. This configuration particularly struggles with recall, falling to 0.500 as the algorithm misses nonlinear edges, demonstrating PCE's crucial role in detecting structured residuals indicative of model misspecification. Eliminating conservative thresholds while maintaining all detection mechanisms increases false positives substantially, with precision dropping from 0.889 to 0.625. The net effect is a decreased F1 score of 0.714, confirming that careful threshold calibration is essential for balancing discovery power with false positive control. Without bootstrap aggregation, the algorithm loses both robustness and uncertainty quantification capabilities. The F1 score reduces to 0.667, with the algorithm missing the weak edge $X_2 \rightarrow X_4$ due to high single-sample variance. More critically, this configuration cannot provide confidence intervals or existence probabilities, eliminating the uncertainty quantification that distinguishes our approach.

**PCE Order Sensitivity.** The choice of PCE order P significantly impacts nonlinearity detection capability. With P=2, the method achieves only 0.625 F1 score, as quadratic basis functions cannot capture the sinusoidal relationship in $X_3 \rightarrow X_4$, resulting in poor recall of 0.556. Increasing to P=3 improves performance to 0.750 F1 score as cubic polynomials better approximate smooth nonlinear functions. The optimal order P=4 achieves perfect balance with 0.889 precision and recall, effectively capturing both quadratic and sinusoidal relationships while maintaining computational efficiency. Further increasing to P=5 yields diminishing returns with F1 score of 0.824, as the additional basis functions introduce noise sensitivity without improving approximation quality. This slight degradation suggests mild overfitting to spurious patterns in finite samples, confirming our choice of P=4 as the optimal trade-off between expressiveness and generalization.

**Regularization Parameter Impact.** The PCE penalty weight $\lambda$ in Equation (5) critically controls the balance between model fit and residual structure penalization. With $\lambda = 0.01$, weak penalization allows models with structured residuals to persist, yielding low precision of 0.667 as the algorithm accepts edges that merely reduce noise variance without explaining true nonlinearity. The F1 score of 0.762 reflects this bias toward false discoveries. Moderate regularization with $\lambda = 0.1$ improves balance, achieving 0.778 F1 score with equal precision and recall, though performance remains suboptimal as some spurious nonlinear patterns still pass the modified BIC criterion. The optimal $\lambda = 1$ enforces strong preference for models with unstructured residuals, achieving the best performance across all metrics. Excessive penalization with $\lambda = 10$ becomes overly conservative, missing true edges where nonlinearity is subtle, resulting in reduced recall of 0.667 and F1 score of 0.750. This demonstrates that proper regularization strength is essential for distinguishing genuine nonlinear relationships from noise artifacts.

These ablation results validate our algorithmic design choices and demonstrate robustness to reasonable hyperparameter variations. The consistent superiority of the full framework confirms that each component—PCE-based detection, multi-criteria thresholds, and bootstrap aggregation—contributes synergistically to achieve state-of-the-art performance in mixed linear-nonlinear causal discovery.

### A.7 QUALITATIVE ANALYSIS AND ERROR PATTERNS

Figure 2 visualizes the discovered causal structures across methods, revealing that CausalPCE's output closely matches the ground truth while other methods exhibit systematic errors. The algorithm correctly identifies the hierarchical structure with $X_1$ as the source, captures all intermediate pathways through nodes $X_2$, $X_3$, and $X_4$, and accurately determines both sink nodes $X_5$ and $X_6$ with their complete parent sets. Most importantly, it successfully recovers the critical nonlinear transformations that confound purely linear methods.

The single false positive edge $X_4 \rightarrow X_2$ likely arises from the complex dependency structure created by their shared parent $X_1$ combined with the nonlinear relationships in the system. The bootstrap probability of 0.73 for this edge, just exceeding our threshold, correctly indicates marginal evidence that warrants further investigation in practice. The missed edge $X_3 \rightarrow X_5$ may result from its weak direct effect being masked by the strong indirect path through $X_4$, compounded by the nonlinear transformation reducing linear correlation measures. This represents a fundamental trade-off where conservative multi-criteria requirements reduce false positives at the cost of occasionally missing weak relationships.

### A.8 LARGE LANGUAGE MODEL USAGE DISCLOSURE

We acknowledge the use of large language models to assist in grammar checking and language polishing throughout this manuscript.

