# OpenReview forum: "Learning Causal Structures from Mixed Dynamics via Polynomial Chaos Expansion"
_ICLR.cc/2026/Conference — ICLR 2026 Conference Withdrawn Submission_

### Official Review · Reviewer_22Y9 · 2025-10-29

**Soundness:** 2
**Presentation:** 2
**Contribution:** 3
**Rating:** 6
**Confidence:** 3

**Summary:**

The authors consider causal inference in settings where we have both linear and nonlinear relationships. They propose an algorithm that leverages polynomial chaos expansion, among others, to find a linear core structure and remaining nonlinear relationships, which are assumed to be sparse. The proposed algorithm performs well on an industrial dataset.

**Strengths:**

1) The problem itself seems relevant and indeed not very well researched.

2) The usage of polynomial chaos expansion is innovative.

3) Evaluation and comparison seem convincing.

**Weaknesses:**

1) I think the claim that $\epsilon_j$ can follow arbitrary distributions is a bit bold, given that Assumption 2 states that it can actually not be all that arbitrary. The noise distribution needs to be such that it allows for identifiability.

2) The paper goes over some details that would be important to fully understand the algorithm. For instance, what exactly are the compressed sensing techniques that you use? Where does $f$ in (11) come from? If things are explained in the appendix, this should be mentioned. This also holds for the proofs of theorems, which are indeed in the appendix.

3) The score (10) is not very well motivated. Overall, it is not that clear to me how exactly the algorithm now supports causal inference in systems with linear and nonlinear dependencies.

4) Bootstrap methods are in the introduction criticised for not being well-understood, but in the end, used for getting edge probabilities. This seems inconsistent and is not explained.

5) The dataset used for the evaluation is only briefly introduced without much detail.

**Questions:**

1) The random variables $Y$ for PCE need to have finite second moment. Does the same hold then also for the variables considered here? This would mean another assumption on the noise distribution.

2) What is a "standard random variable?" Should it be a standard normal random variable? ($\xi$, introduced after (3)).

3) What are the compressed sensing techniques that you use?

4) Can you motivate how your algorithm, and especially the score function (10), supports making causal inference in the considered mixed dynamics setting?

5) Where does $f$ in (11) and (12) come from? What is its structure, and how did we find it?

6) Can you provide more details on the dataset? Will it be made public? What about the code?

---

### Official Review · Reviewer_JyuS · 2025-10-30

**Soundness:** 2
**Presentation:** 2
**Contribution:** 2
**Rating:** 2
**Confidence:** 3

**Summary:**

CausalPCE tackles causal discovery with mixed linear–nonlinear relations. It first learns a sparse
linear scaffold via a GES-style search scored by a PCE-augmented BIC, then probes candidate
nonlinear edges using a simple “two-of-three” test (polynomial LRT, conditional MI, and a
PCE residual index), and finally reports bootstrap edge probabilities and Sobol indices. Claims
include asymptotic consistency under identifiability and sparsity assumptions and polynomial
complexity with fixed PCE order.

**Strengths:**

• Using Polynomial Chaos Expansion on residuals produces a measurable and testable “nonlinearity
index” for model misspecification.

• The two-of-three screening (polynomial LRT, kNN-CMI, PCE reduction) may be easy to
implement.

• The ablation table shows the method degrades in sensible ways when components are removed
and that performance depends predictably on PCE order, which is at least informative
about the design’s levers.

**Weaknesses:**

• No Related Work section. The literature is folded into a short introduction, which does
not meet ICLR expectations and makes novelty hard to judge. Recent ML work on causal
discovery is largely missing, including differentiable DAG search, nonlinear SEMs, kernel
based tests, and uncertainty methods.

• The paper assumes a DAG model but does not explain why this is appropriate. It does not
discuss feedback or cycles. Faithfulness is mentioned but not defined. Basic graph properties
are also unstated.

• Assumption 1 limits the nonlinear part to order O(d) but does not restrict the total number
of edges in the graph. Without a bound on the total number of edges, sparsity is undefined,
and complexity or consistency statements later in the paper have no clear basis.

• For Assumption 2, proper references for the “established identifiability theory” are needed.

• In Assumption 3, the paper mentions “confounders” but never defines what is meant or
how they are handled. It is unclear whether confounders refer to latent common causes,
correlated noise terms, or observed control variables.

• The paper does not justify the index or define it precisely. There is no calibrated test under
a linear null and no explanation of what the index measures or how it grows with signal
strength. Robustness to non Gaussian and heteroskedastic noise is not addressed.

• Many symbols are used without definition, which makes the paper hard to read and verify.
Examples include, e.g., c, log n, L, NPCE, ˆI, MEC(G).

• The experiments rely on an unnamed “real-world industrial process” with known ground
truth, but the paper provides no dataset name, citation, variable descriptions, or access
details. Figures and tables use generic X1–X6 labels, and no provenance is given for the
claimed ground truth, which makes the results non-verifiable and indistinguishable from a
synthetic toy example.

**Questions:**

1. Add a proper Related Work section that situates PCE-on-residuals against modern methods,
states what is new beyond a GES-style scaffold plus residual tests, and justifies the
comparison set.

2. Please say whether the graph can be disconnected, whether there is a bound on in degree,
and whether a topological order is used in the search.

3. Define every symbol on first use and keep names consistent across text, equations, and
algorithms.

4. Please identify and cite the dataset (or release it), describe variables and preprocessing,
and justify the ground-truth graph.

5. Please add standard public benchmarks and at least one named industrial benchmark.
For public data, for example, consider Sachs (Causal discovery with interventional data)
and ALARM (A Logical Alarm Reduction Mechanism). For an industrial setting, consider
the Tennessee Eastman process with metrics against strong baselines.

---

### Official Review · Reviewer_ShKR · 2025-11-01

**Soundness:** 2
**Presentation:** 3
**Contribution:** 2
**Rating:** 2
**Confidence:** 3

**Summary:**

This paper introduces a multi-step algorithm for causal structure learning when structural equations are a mix of linear and nonlinear functions. Methods rely heavily on the polynomial chaos expansion (PCE), which helps quantify nonlinearities in residuals via a "nonlinearity index." The proposed algorithm proceeds in three steps: (i) a modified GES search, (ii) a multi-criterion decision rule to add new edges, and (iii) uncertainty quantification. Consistency is shown, and experimental results on a real-world industrial process show promising results.

**Strengths:**

- Mixed linear/nonlinear relationships pose an interesting and realistic problem that's seldom addressed directly in the literature.

- Efficient uncertainty quantification over edges is an important problem, with somewhat lacking state-of-the-art.

- The paper is generally well-written. Motivations and background materials are quite clear, and the methodology of the proposed algorithm is easy to follow.

- Results on a real-world industrial process are promising, achieving the best SHD and F1 score over all baselines.

**Weaknesses:**

- I'm not so sure what is meant exactly in Assumption 1, where the number of nonlinear relationships is O(d). This is perplexing since standard Big-O notation is asymptotic. If this is to mean that there are at most d nonlinear relationships, then Assumptions 1 and 2 concurrently holding with Gaussian noise is quite difficult, as the standard identifiability assumes all functions are strictly nonlinear.

- There are lots of moving parts in the proposed algorithm, with limited ablation. This is not necessarily bad, but leaves lots of spaces where there is potential room for improvement (e.g., why use the k-nearest neighbor MI estimate? Why stop with three tests? Why are the three tests chosen particularly good? What happens if we use more modern alternatives to GES? Why require 2 tests to pass instead of 1 or 3? etc.).

- The use of GES inherits some rough scalability issues. It would be nice to identify other "step 1" algorithms that could scale more readily with the number of nodes.

- Some modern baselines are missing. For example, Dagma [1] is a quite popular alternative to NOTEARS, and is typically faster and more accurate.

- The breadth of experiments is quite lacking for an ICLR submission, in my opinion. I would expect results on other various popular benchmark datasets. I would also expect to see results, e.g., in purely linear or purely nonlinear scenarios, and how results break down as assumption 1 is pushed.

- As a minor point, (1) is taken to be a "general nonlinear structural equation model" but assumes additive noise.

- As another minor point, there is a missing reference in line 633.

[1] Bello, K., Aragam, B., & Ravikumar, P. (2022). Dagma: Learning dags via m-matrices and a log-determinant acyclicity characterization. Advances in Neural Information Processing Systems, 35, 8226-8239.

**Questions:**

1. What is performance like on a more general set of datasets? What about some more modern alternative CD algorithms? This is the most important question to me.

2. How can acyclicity be strictly enforced? Notice that in the presented results (Figure 2), CausalPCE returns a cyclic graph (I suppose due to step 3).

3. Can cubic complexity in $d$ be avoided with an alternative step 1 algorithm?

4. What exactly is meant in Assumption 1?

---

### Note · Authors · 2025-11-12

I have read and agree with the venue's withdrawal policy on behalf of myself and my co-authors.